# Emergence of NDM-7-Producing *Klebsiella quasipneumoniae* subs. *simillipneumoniae* ST138 in a Hospital from the Northern Region of Brazil

**DOI:** 10.3390/microorganisms13020314

**Published:** 2025-02-01

**Authors:** Amália R. F. Lobato, Mikhail J. S. Souza, Emanoele S. Pereira, Thalyta B. Cazuza, Artur Silva, Rafael A. Baraúna, Danielle M. Brasiliense

**Affiliations:** 1Bacteriology and Mycology Section, Evandro Chagas Institute, Secretariat for Health and Environmental Surveillance, Ministry of Health, Ananindeua 67030-000, Brazil; amalialobato007@gmail.com (A.R.F.L.); mikhail.souza@icb.ufpa.br (M.J.S.S.); saraivaemanuele@gmail.com (E.S.P.); thalytacazuza@iec.gov.br (T.B.C.); 2Biological Engineering Laboratory, Innovation Space, Guamá Science and Technology Park, Belém 66075-750, Brazil; arturluizdasilva@gmail.com (A.S.); r.a.barauna@gmail.com (R.A.B.)

**Keywords:** antibiotic resistance, bacterial genome, carbapenem-resistant, *Klebsiella quasipneumoniae*, New Delhi metallo-beta-lactamase

## Abstract

Clinical emergent bacterial pathogens are a great threat to the global health system, chiefly Gram-negative carbapenem-resistant Enterobacterales and the *Klebsiella pneumoniae* species complex. Here, we present the molecular and phenotypic characterization of *Klebsiella quasipneumoniae* subs. *similipneumoniae* IEC57090 strain, belonging to ST138 and showing a multidrug resistance phenotype. The *bla*_NDM-7_ present in one of the two resistance plasmids carried by the isolate, the antibiotic resistance genes *fos*A, *oqx*AB, and *acr*R, and gene mutations on porins *omp*K36 and *omp*K37, both associated with cephalosporin and carbapenem resistance, were detected. Virulence factors such as the clusters of type I and III fimbria, type IV pili genes, and genes associated with the K1 capsule, siderophore production, and multiple mobile genetic elements (MGE) were predicted. The emergence of silent pathogens in clinical environments highlights the importance of active research on new threads that may compromise the last resources of antimicrobials, such as carbapenems, specifically on mobile genetic elements containing carbapenemases in emergent pathogens, which can spread these antimicrobial resistance elements. This study reinforces that molecular biology vigilance can prevent outbreaks and help to better understand antimicrobial resistance and pathogens in clinical environment dynamics.

## 1. Introduction

*Klebsiella pneumoniae* is a pathogen currently related to healthcare-associated infections (HAIs) and antimicrobial resistance (AMR). Two additional species have recently been described in the genus, *Klebsiella variicola* and *Klebsiella quasipneumoniae*, both of which are opportunistic pathogens associated with urinary tract and bloodstream infections [1].

*Klebsiella quasipneumoniae* is classified into the subspecies *K. quasipneumoniae* subs. *quasipneumoniae* and *K. quasipneumoniae* subs. *similipneumoniae* [2]. Several genomic studies have shown that approximately 3.6% up to 32.5% of the isolates classified as *K. pneumoniae* were inaccurately identified and later reclassified as *K. quasipneumoniae*. Therefore, this species circulates silently in hospital units due to its challenging taxonomic classification when using traditional biochemical methods commonly used in clinical laboratories [1,2,3].

Few resistance and virulence mechanisms were initially described for *K. quasipneumoniae*, but recent studies show the opposite [4]. Several types of carbapenemases, such as KPC, NDM, IMP, GES, and OXA-181, as well as the plasmid-mediated colistin resistance enzyme MCR, have been described in clinical isolates of *K. quasipneumoniae* from India, Brazil, Malaysia, Saudi Arabia, United States, and China. Resistance to other antimicrobial classes such as tetracyclines and tigecyclines has also been described [5,6,7,8,9,10,11,12].

In Brazil, during the COVID-19 pandemic, an increase in the prevalence of New-Delhi Metallo-β-lactamase (NDM)-producing Enterobacterales was observed, especially in *K. pneumoniae* isolates [13]. Thus, therapeutic options for the treatment of HAIs associated with this pathogen have become increasingly scarce [13].

In this study, we characterized a multidrug resistant (MDR) NDM-7-producing *Klebsiella quasipneumoniae* subs. *Similipneumoniae,* isolated from an adult patient admitted to the intensive care unit (ICU) of a tertiary hospital in the city of Belém, state of Pará, Brazilian Amazon, during the COVID-19 pandemic. The NDM-7 variant has rarely been reported in Latin America and was recently described in *K. pneumoniae* isolates from hospitals in the northern region of Brazil [14]. To the best of our knowledge, this is the first report of the NDM-7 variant in a *K. quasipneumoniae* isolate in Brazil, demonstrating that plasmids carrying the *bla*_NDM_ gene can be spread among different members of Enterobacterales. Our data reinforce the need for additional studies to prevent the spread of this carbapenemase variant and show that routine practices based on clinical surveillance by active search for MDR isolates carrying these carbapenemase enzymes are extremely important.

## 2. Material and Methods

The IEC57090 isolate was obtained from the peritoneal fluid of an adult patient admitted to the ICU of a 220-bed tertiary hospital in the city of Belém, State of Pará, in November 2020. Initial taxonomic identification was performed at the hospital of origin, where the isolate was classified as *K. pneumoniae* by a Vitek 2 automated system (Biomerieux, Marcy-l’Étolle, France).

The isolate was sent to the Evandro Chagas Institute for antimicrobial resistance surveillance, where the carbapenemase-encoding genes *bla*_NDM_, *bla*_KPC_, *bla*_IMP_, *bla*_VIM_, and *bla*_OXA-48_ were investigated by Polymerase Chain Reaction (PCR). The primers, master mix composition, and cycling conditions have been previously described by Han et al. [15]. The GoTaq™ G2 Flexi DNA Polymerase kit (Promega, Madison, WI, USA) was used for PCR, with the follow conditions: one cycle of 95 °C for 2 min, 35 cycles of amplification (95 °C for 30 s, 52 °for 40 s, 72 °C for 50 s) and one cycle of 72 °C for 10 min. Amplicons were visualized in a 1% agarose gel electrophoresis with SafeDye Nucleic Acid Stain (Cellco, São Paulo, Brazil). Sanger sequencing to verify the NDM allele was conducted with a BigDye™ Terminator v3.1 Cycle Sequencing Kit (Thermo Fisher Scientific, Cleveland, OH, USA) in a 3500xL Genetic Analyzer (Thermo Fisher Scientific, Cleveland, OH, USA).

Antimicrobial susceptibility testing (AST) was performed by agar dilution or broth microdilution methods using Sensititre Gram-negative GNX3F AST CI (TREK Diagnostic Systems, Thermo Fisher Scientific, Cleveland, OH, USA), according to the Clinical and Laboratory Standard Institute (CLSI) and Food and Drug Administration (FDA) criteria for the antimicrobials ampicillin, piperacillin/tazobactam, cefuroxime, ceftazidime, cefepime, aztreonam, imipenem, meropenem, gentamicin, amikacin, tetracycline, ciprofloxacin, colistin, polymyxin, and tigecycline [16,17].

Whole-genome sequencing was performed on the Illumina HiSeq platform using 2 × 150 bp paired-end libraries (Illumina Inc., San Diego, CA, USA). Bacterial DNA was extracted using the NucleoSpin Microbial DNA Kit (Macherey-Nagel, Dueren, Germany), according to the manufacturer’s protocol. A bacterial colony from an agar nutrient plate culture was eluted in 300 μL of ultra-pure water, mixed until homogeneous, and then centrifuged at 13,000× *g*. The supernatant was discarded and 100 μL of elution buffer was added. After mixing, the solution was transferred to a MN Tube Type B for bacterial lysis, and 40 μL of buffer MG and 10 μL of liquid proteinase K were also added. The microtube was agitated for 12 min and centrifuged 30 s at 11,000× *g*. After adjusting the binding conditions, 600 μL of the solution was transferred to the NucleoSpin^®^ Microbial DNA Column. The silica column was washed with 500 μL of BW and 500 μL of B5 solutions, and then centrifuged at 11,000× *g* for 30 s. The column was placed in a final collection microtube and eluted with 50 μL of elution buffer. Raw data were filtered and trimmed using Sickle v.1.33 [18], and bases with less than Phred 20 quality and reads shorter than 40 bases were eliminated. Genome assembly was performed using SPAdes v.3.13.0 [19], and the gaps were closed using GMCloser v.1.5 [20]. Finally, the consensus sequence was generated using AlignGraph v.1 [21] and MeDuSa v.1.6 [22].

Antimicrobial resistance genes (ARGs) were detected using ResFinder v.4.1 [23] and CARD RGI v.6.0.3 [24]. The plasmid sequence was detected using PlasmidFinder v.2.1 [25] and MOB-Suite v.3.1.4 [26]. Virulence factors were predicted using VFanalyzer [27], and Multi-Locus Sequence Typing (MLST) analysis was achieved using PathogenWatch (http://www.pathogen.watch, 5 May 2023). Mobile Genetic Elements (MGEs), Genomic Islands (GIs), and prophages were predicted using MGEFinder v.1.0.3 IslandViewer v.4 and PHASTER (http://www.phaster.ca, 25 January 2024), respectively [28,29,30].

## 3. Results and Discussion

Antimicrobial susceptibility testing of the IEC57090 isolate revealed high values of minimum inhibitory concentrations (MIC) for several drugs, including beta-lactams such as ampicillin, ampicillin/sulbactam, piperacillin/tazobactam, cefuroxime, ceftazidime, cefepime, imipenem, meropenem, doripenem, and intermediate to doxycycline. The strain showed in vitro susceptibility to aztreonam, gentamicin, amikacin, tobramycin, tetracycline, ciprofloxacin, levofloxacin, minocycline, tigecycline, sulfamethoxazole/trimethoprim, polymyxin B, and colistin. Based on its resistance profile, the isolate was classified as MDR (Table 1) [31]. The *bla*_NDM_ gene was detected by PCR and subjected to Sanger sequencing, being classified into the NDM-7 variant by alignment with the 47 *bla*_NDM_ alleles from CARD [24] on BioEdit v.7.7.

Subsequently, the whole genome of the IEC57090 isolate was sequenced and assembled into a consensus sequence of 5.5 Mbp, with a coverage of approximately 235×, 58.73% of GC content, and 5.230 coding sequences (CDSs). Taxonomic classification was performed based on *rpo*B and whole-genome sequence using TYGS v.391 [32] with available reference *Klebsiella* spp. genomes (Appendix A). The genome sequence was deposited in GenBank under the accession number CP178200.

The taxonomic classification performed by TYGS based on phylogenomic analysis classified the IEC57090 strain as *K. quasipneumoniae* subs. *similipneumoniae* (Figure 1). In contrast, the phylogenetic tree calculated based on the *rpo*B gene sequence alone did not provide sufficient resolution to distinguish the species of the *K. pneumoniae* species complex. Thus, phylogenomic and Average Nucleotide Identity (ANI) analyses of whole-genome sequences were the most effective methods for taxonomic classification [2,33].

Two plasmids were detected in the genome of the IEC57090 isolate belonging to the IncFIIB and IncX3 incompatibility groups, with average sizes of 176 kbp and 44 kbp, respectively. The pIEC57090-FIIB plasmid has 175,805 bp, 180 CDSs, and showed 99.96% identity with the pA2508-emrE plasmid from a *K. quasipneumoniae* isolate (accession number MN310379.1). Among the main genes found in pIEC57090-FIIB, we highlight the *tra*AEDILMJXY conjugation system, the toxin–antitoxin (TA) systems *vap*BC and *hig*AB, the copper resistance determinant *pco*, and resistance to quaternary ammonium compounds gene *qac*E.

Regarding plasmid pIEC57090-X3, from 44 kbp size, 58 CDS were predicted, of which the main genes found were *bla*_NDM-7_ and the type IV secretion system *vir*B (Figure 2). The *bla*_NDM-7_ gene was embedded in a Tn*125*-like element that was composed of the structure *dsb*D-*trp*F-*ble*_MBL_-*bla*_NDM-7_-IS5-IS*Aba125*-IS*3000*-*hin*. The same structure was observed in the pYUSBH035 plasmid (accession number LC716358.1), which showed 99.99% identity and 93% coverage with pIEC57090-X3 and was detected in a *K. quasipneumoniae* ST 196 strain from a patient in China [34].

MLST analysis assigned the isolate to IEC57090 belonging to the sequence type ST138, an undefined clonal complex in PubMLST (http://www.pubmlst.org, 5 May 2023), composed of *bla*_GES_-producing *K. michiganensis* isolates [35], *bla*_GES-5_-producing *K. quasipneumoniae* isolates from a wastewater treatment plant in Slovenia [36], and other *K. quasipneumoniae* strains with multiple mechanisms of AMR and virulence that were isolated from an international space station [37]. Thus, this is the first report of a *bla*_NDM_-harboring IncX3 plasmid in an isolate belonging to ST138.

Acman and colleagues (2022) examined 6000 bacterial genomes harboring the *bla*_NDM_ gene and described the IncFII and IncX3 incompatibility groups as the main carriers of this gene [38]. Interestingly, both incompatibility groups’ plasmids were detected in the genome of *K. quasipneumoniae* IEC57090 strain. Although the pIEC57090-FIIB plasmid does not carry any resistance genes, it has 14 insertion sequences in its structure, making it possible to acquire the *bla*_NDM_ gene in recombination events.

**Figure 2 microorganisms-13-00314-f002:**
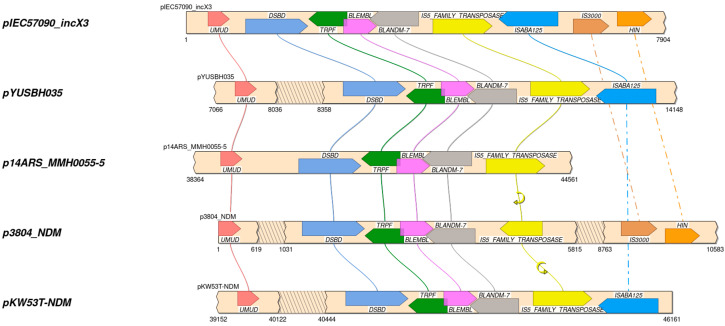
Synteny of the *bla*_NDM_-harboring MGE found in the pIEC57090-X3 plasmid. The sequence was compared with the pYUSBH035, p14ARS_MMH0055-5, p3804_NDM, and pKW53T-NDM plasmids, which were found to be the most closely related to pIEC57090-X3 according to BLASTn analysis. The figure was drawn using SimpleSyteny v.1.6 [39].

In addition to the *bla*_NDM-7_ gene, resistome analysis also revealed the presence of *fos*A (fosfomycin). The outer membrane porins *omp*K36 and *omp*K37, both associated with cephalosporin and carbapenem resistance, were also detected. The viruloma analysis revealed the presence of complete clusters of type I and III fimbria, and type IV pili genes. It is worth noting that type III fimbria have been described as the major adhesion factor in *K. quasipneumoniae* [40], in contrast to *K. pneumoniae*, which preferentially express the type I gene cluster.

Nine families of insertion sequences were found in the *K. quasipneumoniae* IEC57090 genome (*IS5, ISKpn19, ISKpn21, ISKox3, ISSen4, ISKpn26, ISEhe3, ISEhe3, ISEam1*) and five intact prophages containing CDSs encoding transposases, endonucleases, integrases, and endolysins in addition to typical phage proteins. Twenty-six GIs were predicted to harbor multiple insertion sequences, a type IV secretion system, endonucleases, an ABC-type ferric iron transporter, a *cus* efflux system, and the quinolone resistance gene *nor*B. Some prophages were found to have overlapping GI sequences, such as the region encoding the *ter*L endonuclease subunit, an enzyme related to viral translocation [41]. The detailed results are presented in Appendix A.

Our results raise that the dissemination of carbapenemase genes among species of the *Klebsiella pneumoniae* complex is probably underestimated. Low-resolution methods of bacterial identification have also led to misidentifications, underestimating other species in the complex. In addition, the selective pressure of the hospital environment promotes the spread of carbapenemases among phylogenetically related pathogenic bacteria [2]. Mathers and colleagues (2019) demonstrated that *K. quasipneumoniae* was susceptible to acquiring plasmids from other Enterobacterales [9] a finding of public health importance because some bacterial species may serve as sentinels for the presence of MGEs carrying ARGs and virulence factors in hospitals. Under selective pressure, these genes can be transferred to well-established pathogens, increasing public health expenditures and impacting population health.

## 4. Conclusions

In our study, we characterized the genome of a multidrug-resistant and carbapenemase-producing *K. quasipneumoniae* subs. *similipneumoniae* IEC57090 strain from a hospital in the Amazon region, Brazil. Our data showed that in addition to *K. pneumoniae*, other members of the *K. pneumoniae* complex are silently circulating in hospitals in the northern region. The IEC57090 isolate presented several ARGs, MGEs, virulence factors, and two plasmids belonging to incompatibility groups commonly related to antimicrobial resistance gene carriers. Thus, our study highlights the importance of using genomic tools in epidemiologic studies for AMR surveillance.

## Figures and Tables

**Figure 1 microorganisms-13-00314-f001:**
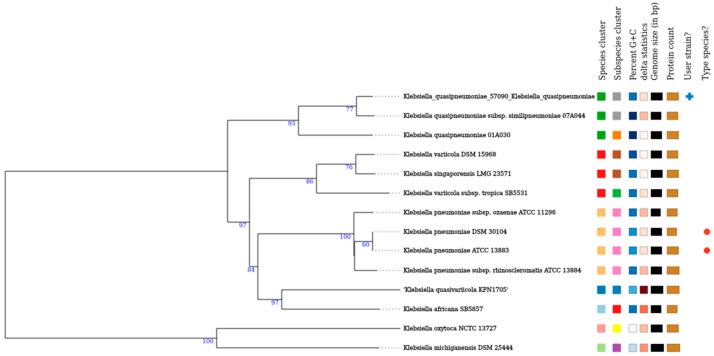
Taxonomic classification of the isolate IEC57090. The phylogenomic tree was calculated in the Type Strain Genome Server, based on the core genome of isolate IEC57090 and other members of the *K. pneumoniae* species complex. Bootstrap values for 100 replicates are shown in blue next to each branch.

**Table 1 microorganisms-13-00314-t001:** Antimicrobial susceptibility profile of *K. quasipneumoniae* IEC57090 strain.

Antimicrobial	MIC (µg/mL)	Interpretation
Ampicillin	>256	R
Ampicilin/sulbactam	>256/4	R
Piperacillin/tazobactam	>256/4	R
Cefuroxime	>256	R
Ceftazidime	>256	R
Cefepime	256	R
Aztreonam	≤1	S
Imipenem	32	R
Meropenem	64	R
Doripenem	>4	R
Gentamicin	≤1	S
Amikacin	4	S
Tobramycin	2	S
Tetracycline	2	S
Ciprofloxacin	2	S
Levofloxacin	≤1	S
Doxycyclin	8	I
Minocycline	4	S
Tigecycline	0.5	S
Sulfamethoxazole/trimethoprim	≤0.5/9.5	S
Colistin	≤0.25	S
Polymyxin B	≤0.25	S

R: resistant, I: intermediate, and S: susceptible.

## Data Availability

The original contributions presented in the study are included in the article/Appendix A; nucleotide data present in this study are available in NCBI biosample, accession SAMN44590899. Further inquiries can be directed to the corresponding author.

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
