# Peer review of "Emergence of NDM-7-Producing Klebsiella quasipneumoniae subs. simillipneumoniae ST138 in a Hospital from the Northern Region of Brazil"

_microorganisms, 2025, doi:10.3390/microorganisms13020314_

Round 1

Reviewer 1 Report

Comments and Suggestions for Authors

Antimicrobial resistance is an emerging worldwide problem. The surveillance of resistant strains is highly important to prevent the outbreaks. Molecular biology vigilance is an extremely useful tool to characterize and detect resistant isolates and support the fight against the antimicrobial resistance. In this manuscript, the authors present the phenotype and molecular characteristics of Klebsiella quasipneumoniae, simillipneumoniae isolate that present multidrug resistance. The report is comprehensive , well -structured and well- presented. Thus,  I accept this manuscript for publication after the implementation of one minor comment:

Lines 38-40. The authors have to rephase the sentence. It is not clear what the authors want to state.

Author Response

Comments 1: Lines 38-40. The authors have to rephase the sentence. It is not clear what the authors want to state.

Response 1: We agree with the comment. Therefore, we refrased the lines between 38-40 and updated in the text manuscript. From: "Klebsiella quasipneumoniae is divided into the subspecies K. quasipneumoniae subs. quasipneumoniae and K. quasipneumoniae subs. similipneumoniae (Long et al., 2017). Several genomic studies have shown that approximately 3.6% up to 32.5% of the isolates classified as K. pneumoniae were reclassified as K. quasipneumoniae."

To: "Klebsiella quasipneumoniae is classified into the subspecies K. quasipneumoniae subs. quasipneumoniae and K. quasipneumoniae subs. similipneumoniae (Long et al., 2017). Several genomic studies have shown that approximately 3.6% up to 32.5% of the isolates classified as K. pneumoniae were inaccurately identified and after reclassified as K. quasipneumoniae."

Reviewer 2 Report

Comments and Suggestions for Authors

Article title: Emergence of NDM-7-producing Klebsiella quasipneumoniae subs. simillipneumoniae ST138 in a hospital from the Northern Region of Brazil 

The article puts into perspective a problem that is becoming more and more severe: the emergence of new and resistant bacterial strains. There are currently very few articles describing Klebsiella quasipneumoniae subs. simillipneumoniae published in literature and even less describing the strain studied by the authors. Therefore, I consider this communication could fill a gap in literature by characterizing this emerging strain.

I only have some minor suggestions regarding the Material and methods section:

- line 77 - please modify "Polymerase ChainHAN Reaction" to "Polymerase Chain Reaction"

- line 78 - can you please briefly describe in the text the protocol/primers mentioned (Han et al.)?

- line 92 - please also briefly describe the protocol used for DNA extraction in the manuscript text

I also have a curiosity. In the first paragraph of the Material and methods section, it is mentioned that initial taxonomic identification was performed using Vitek 2 automated system, and the strain was (wrongly) identified as Klebsiella pneumoniae, and only after genetic analysis in a reference laboratory was the strain actually identified as Klebsiella quasipneumoniae subs. simillipneumoniae. I think this raises a very interesting concern, which you also discussed really well in the manuscript: the strain characterized here might be more common than we think! I was wondering if you tried other means of Automatic identification (e.g. MALDI-TOF) and if these methods managed to identify the strain correctly or not? If you did perform any other tests for identification, I think it would be interesting to mention this in the article as well. 

Author Response

Comments 1:- line 77 - please modify "Polymerase ChainHAN Reaction" to "Polymerase Chain Reaction"
Response 1: We agree with this comment. Corrected from "ChainHAN" to "Chain".
Comments 2: - line 78 - can you please briefly describe in the text the protocol/primers mentioned (Han et al.)?
Response 2: We agree with this comment and the protocol has been described as "The GoTaq™ G2 Flexi DNA Polymerase kit (Promega, Madison, Wisconsin, United States) was used for PCR with the follow conditions: one cycle of 95° C for 2 min, 35 cycles of amplification ( 95° C for 30 s, 52° C for 40 s, 72° C for 50 s) and one cycle of 72° C for 10 min. Amplicons were visualized in a 1% agarose gel electrophoresis with SafeDye Nucleic Acid Stain (Cellco, São Paulo, Brazil)."
Comments 3: - line 92 - please also briefly describe the protocol used for DNA extraction in the manuscript text
Response 3: We agree with this comment and the protocol has been described as "A bacterial colony from an agar nutrient plate culture was eluted in 300 μL of ultra pure water, mixed until homogenity and then centrifuged at 13,000 x g. Supernatant was discarded and 100 μL elution buffer was added. After mixing, the solution was transferred to a MN Tube Type B for bacterial lysis and 40 μL of buffer MG and 10 μL of liquid proteinase K were also added. The microtube was agitated for 12 min and centrifuged 30 s at 11,000 g. After adjusting binding conditions, 600 μL of the solution was tranferred to the NucleoSpin® Microbial DNA Column. The silica column was washed with 500 μL of BW and 500 μL of B5 solutions, then centrifuged at 11,000 x g for 30 s. The column was placed in final collection microtube and eluted with 50 μL of elution buffer."
Comments 4: I was wondering if you tried other means of Automatic identification (e.g. MALDI-TOF) and if these methods managed to identify the strain correctly or not? If you did perform any other tests for identification, I think it would be interesting to mention this in the article as well."
Response 4: About the identification of the species in question, we did not perform any other tests for identification besides the described in the manuscript, but currently the best method for identification of Klebsiella quasipneumoniae is whole genome sequencing or sequencing of house-keeping gene rpoB. Chen, Long, Octavia and respective co-authors embase the affirmation and can be found in the manuscript’s references. Rodrigues et al. in 2018 have proposed spectra of all K. pneumoniae complex members for MALDI-TOF identification, although it was not  incorporated in the spectra databases, which difficult correct identification of the species by the cited method (Article doi: 10.3389/fmicb.2018.03000). We are very greatful for all revisions.